# CD99 and the Chicken Alloantigen D Blood System [note 1]

**DOI:** 10.3390/genes14020402

**Published:** 2023-02-03

**Authors:** Janet E. Fulton, Wiola Drobik-Czwarno, Ashlee R. Lund, Carl J. Schmidt, Robert L. Taylor

**Affiliations:** 1Hy-Line International, Dallas Center, IA 50063, USA; 2Department of Animal Genetics and Conservation, Institute of Animal Science, Warsaw University of Life Sciences, 02-787 Warsaw, Poland; 3Department of Animal and Food Science, University of Delaware, Newark, DE 19716, USA; 4Division of Animal and Nutritional Sciences, West Virginia University, Morgantown, WV 26506, USA

**Keywords:** Chicken D blood system, CD99, Xg blood system, chicken–human synteny

## Abstract

The chicken D blood system is one of 13 alloantigen systems found on chicken red blood cells. Classical recombinant studies located the D blood system on chicken chromosome 1, but the candidate gene was unknown. Multiple resources were utilized to identify the chicken D system candidate gene, including genome sequence information from both research and elite egg production lines for which D system alloantigen alleles were reported, and DNA from both pedigree and non-pedigree samples with known D alleles. Genome-wide association analyses using a 600 K or a 54 K SNP chip plus DNA from independent samples identified a strong peak on chicken chromosome 1 at 125–131 Mb (GRCg6a). Cell surface expression and the presence of exonic non-synonymous SNP were used to identify the candidate gene. The chicken CD99 gene showed the co-segregation of SNP-defined haplotypes and serologically defined D blood system alleles. The CD99 protein mediates multiple cellular processes including leukocyte migration, T-cell adhesion, and transmembrane protein transport, affecting peripheral immune responses. The corresponding human gene is found syntenic to the pseudoautosomal region 1 of human X and Y chromosomes. Phylogenetic analyses show that CD99 has a paralog, XG, that arose by duplication in the last common ancestor of the amniotes.

## 1. Introduction

In humans, CD99 is a cell surface protein, involved with multiple biological processes including cell adhesion, migration, and cell death. It is essential for the transendothelial migration of immune cells. It plays an important role in lymphocyte development, is involved with the upregulation of MHC class I and II and T cell receptor expression on thymocytes, and has important roles in both T-cell activation and early B-cell lymphopoiesis [1]. The CD99 protein, together with its highly conserved homolog Xg antigen, results in the Xg blood group [2]. CD99 is expressed ubiquitously at low levels, whereas Xg is expressed solely on erythrocytes [3].

Human Xg and CD99 genes, each consisting of 10 or 11 exons (alternative splice site), encode the Xg^a^ and CD99 glycoproteins, respectively, which are separately expressed on the red blood cell (RBC) membrane. They are located adjacent to each other in the genome in the same orientation. In humans, the CD99 gene lies within pseudoautosomal region 1, while part of the Xg gene is located within the pseudoautosomal region and a portion is X-specific. Thus, in humans, an incomplete Xg is present on the Y chromosome. The expression of Xg and CD99 is coregulated and these two molecules together define the human Xg^a^/CD99 blood group [4].

An increased understanding of how variability within genes affecting immune responses can impact survival is relevant to improved animal health. The chicken is an excellent non-mammalian model organism for comparative immunology as well as an important food source. Early immunology studies revealed the partitioning of B- and T-cell development [5,6]. Rous sarcoma virus, the first tumor virus, and SRC, the first oncogene, were isolated in the chicken [7]. An increased demand for food due to a rising population stresses the importance of the chicken’s high-quality meat and egg protein. Innate and acquired immune responses are becoming more important due to the declining availability of antibiotics for animal production systems.

There are 13 blood systems known in the chicken which were named alphabetically in their order of discovery [8]. The B blood group system is now known to be the Major Histocompatibility Complex (MHC) which contains multiple immune-related genes and has a major impact on disease resistance [9,10]. The genes responsible for the A and E blood systems have recently been identified as C4BPM and FCAMR, respectively, with both genes having significant roles in the immune response [11]. Very little information on candidate gene or genomic location is known for any of the other chicken blood systems.

The availability of DNA banks comprising multiple DNA samples from individuals serologically typed for various blood systems, plus the availability of genome sequence information for several experimental lines with known blood system alleles, provided the resources for the identification of candidate genes of chicken blood systems. The chicken D blood system was first identified by Briles as the fourth red blood cell antigen [12]. The D system was autosomal with no linkage reported to any other known alloantigen or to multiple phenotypic traits including naked neck (Na) and pea comb (P) [9]. Classical recombination studies utilizing morphological phenotypes, and a chromosome rearrangement breakpoint located the D system gene on chromosome 1 [13].

Information on the biological role of the D system in the chicken is scarce; however, the few available studies point to a role in immune response. Divergent selection for bursa of Fabricius size or antibody response against sheep red blood cells in chicken lines resulted in differences in the frequency of D alleles indicating possible relationships with the D system [14,15,16]. Lipopolysaccharide (LPS) stimulation of macrophages triggered a higher nitrite production in D^1^D^2^ chickens compared with D^1^D^3^, D^1^D^4^, D^2^D^3^, and D^3^D^3^ genotypes within specific MHC-B genotypes [9]. However, no other D system effects have been reported to date.

Little research has been carried out with the D blood system since these early studies, most likely due to lack of alloantisera for the detection of the antigens. The primary purpose of the research presented herein was to determine the gene responsible for the D blood system antigen in the chicken. Identification of the causative gene, and DNA-based detection of the variants defining the D alleles would provide the tools needed to expand our information on the D system’s relevance to the chicken immune response. The observation that the candidate gene has an immunological impact on the mammalian immune system adds considerable value to the identification of the chicken D blood system antigen.

## 2. Materials and Methods

### 2.1. Genetic Material

DNA was available from birds from multiple sources with serologically defined D system alleles and is summarized in Table 1. The Northern Illinois University (NIU) DNA bank contained a set of pedigree samples produced from 2 sires (D^1^D^2^) each mated to a single dam (D^1^D^2^), with serological information for the segregating progeny. In addition, there were NIU non-pedigree samples having known D^1^ and/or D^2^ alleles. The HAS/LAS lines are a pair of lines divergently selected for high (HAS) or low (LAS) antibody levels after immunization with sheep red blood cells. Generations 11 and 13 had previously been typed for the D system and were reported as containing D^1^, D^2^, and D^4^ [15]. DNA was available for generation 41. UCD001 is the inbred Red Jungle Fowl line used as the source of the chicken genome reference (up to GRCg6a), and while its D system allele is not reported, it is likely to be similar to UCD003 [17] as this is not one of the 6 blood system alleles utilized for mapping. Lines WL1–5 are elite White Leghorn (WL) lines from Hy-Line International (Dallas Center, IA, USA) for which multiple generations of individuals with serologically defined D system allele information were available, either as segregating individuals or lines identified as homozygous for a specific allele. Lines RIR1 and WPR1–2 are elite lines from Hy-Line International and are Rhode Island Red (RIR) and White Plymouth Rock (WPR) breeds, respectively.

DNA pools were produced using the two NIU pedigree families for which both sire and dam were heterozygous D^1^/D^2^. Three DNA pools were prepared for each family, containing DNA of either D^1^D^1^, D^1^D^2^, or D^2^D^2^ alleles as determined by serology (3–8 DNA per pool, total sample size of 28). An additional set of DNA pools was generated from NIU non-pedigree birds, and was comprised of either D^1^D^1^, D^1^D^2^, or D^2^D^2^ individuals. Genome sequences were available for the five experimental lines listed (courtesy of Jacqueline Smith, Roslin Institute, and Hans Cheng, USDA/ADOL), four of which also had known serologically defined D system alleles. Genome sequences were also available for elite WL, RIR, and WPR layer lines [18].

### 2.2. GWAS Analysis

SNP genotypes for the three sets of DNA pools were obtained using the Affymetrix Axiom 600 K chicken SNP array. Genotyping was performed by GeneSeek (Lincoln, NE, USA). Genotype calling was performed with Affymetrix Analysis Power Tools. A quality control filter was applied with minor allele frequency of 0.1 and maximum missing genotypes of 0.1 resulting in genotypes from 580,961 SNP available for the analysis.

### 2.3. Identification of Candidate Gene

Intensity files from the 600 K SNP Affymetrix chip were exported and processed using a custom R script. Calls were recoded as 0 for AA, 1 for AB, and 2 for BB to represent the number of copies of B allele in the individual. A regression equation was developed for each SNP. Expected allele count (based on serology information) was used as response variable and intensity was used as explanatory variable. Coefficient of determination (*r^2^*) calculated from regression analysis was used to select SNPs that segregated in accordance with expected pool genotypes. All genes on chromosome 1 were examined, with subsequent focus on region with highest *r^2^* value.

BWA and standard GATK pipeline were used to align sequences to GrCg6a and to call variants. The programme SNPEff [22] was used for variant annotation, while the programme SNPSift [23] identified variants with HIGH and MODERATE impact on the protein function according to sequence ontology terms. Biomart [24] and Uniprot databases [25] were used to identify cellular components for all proteins. Only genes whose proteins were known to be expressed on cell surface membranes were considered. Frequencies of SNPs and INDELs with high and moderate impact in the region of interest were compared between inbred and HYL lines fixed or segregating for D alleles. The candidate protein was the only one that was present on the plasma membrane, with missense variants which matched expected frequencies.

### 2.4. SNP Selection and Subsequent Genotyping

SNP within the candidate gene were selected based on whether they were predictive for non-synonymous changes, as protein differences are expected to be one cause of serological differences among blood system alleles. Additional synonymous SNP variants were chosen to provide better coverage of the gene, plus one indel within a potential splice region was also included. SNP allele and indel genotypes were identified through PACE ^®^ (PCR Allelic Competitive Extension) chemistry (3CR Bioscience Ltd., Harlow, UK) which employs one common primer with two allele-specific primers and fluorescence detection with end-point reads [26]. All functional SNP within the candidate gene, their genomic and gene location, and their putative nucleotide change with corresponding amino acid change are listed in Table 2.

Samples from individuals with serologically defined D alleles, or from lines known to contain specific D system alleles were genotyped with the SNP set identified within the candidate gene. Specific combinations of SNP alleles were identified and where possible, SNP haplotypes were compared with D serological alleles, to associate specific haplotypes with specific D alleles. These haplotypes were then assigned an identifier matched with the serological moniker. DNA from RIR and WPR lines were included to determine if additional novel haplotypes were present in these two different breeds even though no serological information was available.

### 2.5. Sequence Comparison Methods

Ortholog and paralog predictions were performed using BLAST [27,28,29]. Synteny analysis was performed using SynSearcher and SynBrowser available from the Synteny Portal [30]. Multiple sequence alignments were performed with COBALT [31] and phylogenetic trees were generated with the interactive Tree of Life web server [32]. Protein topology and secondary structure were predicted using PredictProtein [33] and structural predictions were obtained from AlphaFold [34,35].

## 3. Results and Discussion

### 3.1. Identification of D System Candidate Gene

Results from a GWAS analysis using 600 K chicken SNP array genotypes from the pooled samples containing DNA from either D^1^, D^2^, or heterozygous pedigree individuals from the NIU DNA bank are shown in Figure 1A. A single strong peak was observed on chromosome 1. Closer examination of chromosome 1 (Figure 1B) shows that this narrow peak encompasses the 129.7 Mbp to 131.4 Mbp region. This region contains five genes with GO terms involving integral components of the membrane: ARSD, ARSE, CD99, P2RY8, and SLC25A6. This genomic region was confirmed by GWAS analysis with an independent set of samples (HYL1; *n* = 50) for which the D allele was known (D^2^, D^3^) and 54 K genotypes were available.

With the genomic region identified and independently confirmed, we attempted to narrow down to the specific gene utilizing available genomic sequences. Four inbred lines were known to contain either the D^1^ allele (line IAH-7_2_) or the D^3^ (lines UCD003, IAH-6_1_, and IAH-RHC) (see Table 1). A comparison of the sequences in this region for the D^1^ vs D^3^ lines showed that only those SNP within the CD99 gene fit with the known phenotype pattern, i.e., all D^3^ lines were homozygous and identical and differed from the D^1^ containing line. This identified the most likely candidate gene as CD99.

A close examination of the sequences of HYL lines for which D allele segregation had been previously identified showed that lines WL1, WL3, and WL4 had SNP segregation within CD99, while lines WL2 and WL5 did not. This specific SNP segregation pattern was not seen within any of the other genes within the candidate region identified in the GWAS analyses, including ARSD and ARSE. With the SNP patterns fitting to the expectation based on serological information, these additional sequences confirmed CD99 as the most likely D blood system candidate gene.

### 3.2. Haplotype Identification

The SNP panel developed for the CD99 gene detected 7 SNP and one indel, encompassing exons 2 through 11. Genotypes for each assay were obtained from individuals from the HYL lines for which D system allele information was known. A total of 11 unique CD99 haplotypes were found (see Table 2). The three inbred lines that contained the D^3^ allele were all homozygous for the same CD99 haplotype, which was subsequently labelled CD99-H03. The inbred line with the D^1^ allele contained a different haplotype, labelled as CD99-H01.

Lines WL2 and WL5, both of which are D^3^ homozygous, contained only CD99-H03, thus confirming that D^3^ = CD99-H03. For line WL1, CD99 haplotypes were defined for 58 individuals previously typed as either D^2^ and/or D^3^. Two haplotypes were found, CD99-H03 and another haplotype subsequently named CD99-H02; thus, D^2^ = CD99-H02. There was 100% agreement with the serological D alleles previously assigned and the CD99 haplotypes.

Line WL3 had previously been reported as segregating for D^1^ and D^3^, though no individual DNA with known D alleles was available. Within the 120 individuals tested, there were three CD99 haplotypes found: CD99-H01, CD99-H03, and a rare haplotype (found in two half-sibs) which was assigned the label CD99-H06.

Line WL4 segregated for alleles D^1^ and D^3^ and contained two CD99 haplotypes, CD99-H01 and CD99-H03, as found in the inbred lines and other Hy-Line lines containing these two serological types. Of the 119 samples for which both serological D alleles and CD99 haplotypes were identified, there was agreement for 106/119 (89%). None of these discrepancies involved the misidentification of homozygotes and are likely due to reagents failing to distinguish heterozygotes accurately.

The HAS and LAS lines both contained CD99-H01 and CD99-H03 and HAS also contained the CD99-H06 haplotype. The D blood system typing of generations 10 and 13 identified both lines as containing D^1^ and D^3^, which is consistent with the presence of CD99-H01 and CD99-H03 that we found in the generation 41 samples. Haplotype CD99-H06 is identical to CD99-H03 at all non-synonymous SNP and thus is expected to produce a protein identical to D^3^. The D^4^ allele previously identified within the LAS line has likely been lost during the intervening 28 generations.

There were 104 samples from the NIU DNA bank for which both serological D allele and CD99 haplotype information was available. The D alleles identified included D^1^, D^2^, and D^3^, with the three corresponding CD99 haplotypes also being found. There was agreement between the D allele and CD99 haplotype for 96/104 of the samples (92%). Samples with disparate results always had one allele correct, and the second allele was either not identified, or was mis-identified.

CD99 haplotypes were also obtained from samples for which no D blood system was available. These were the different breeds Rhode Island Red (RIR) and White Plymouth Rock (WPR)) and were examined to determine if additional haplotypes were present in different breeds. Within the RIR1 line, there were three haplotypes found (CD99-H04, CD99-H07 and CD-H08), and the WPR lines contained CD99-H01, CD99-H04, and novel haplotypes (CD99-H10 and CD00-H11). None of these have any serological information and thus cannot be assigned to specific D alleles; however, these two novel types differ from another haplotype by only one synonymous SNP and thus are likely not another D serological allele. All haplotypes found are reported in Table 2. A summary of the D system alleles included in this study and their corresponding CD99 haplotypes is given in Table 3. The source of each allele and haplotype is also provided.

### 3.3. Protein Structure and Antigenicity

Figure 2 shows the predicted structure of the chicken CD99. The signal peptide and transmembrane domains are predicted to be α helices, but the bulk of the molecule is predicted as a random coil. The location of the SNP used to define the chicken CD99 haplotypes is indicated on the figure. Except for the first SNP (V27M) which is on the signal peptide, they are located on the random coil. While these variants are sufficient to induce an antibody response, whether they have any impact on the functionality of the protein is unknown. The extracellular domain likely extends from the end of the signal peptide to the beginning of the transmembrane domain, encompassing the three amino acid variants V37I, G48D, and E124G. The intracellular domain likely extends from the end of the transmembrane domain to the carboxy terminus, and contains no amino acid variants.

### 3.4. Synteny Analysis

Blast (28, 29), multiple sequence alignments, and synteny analysis were used to further explore the relationships between CD99 and XG. Blast was used to identify CD99 and XG orthologs and a multiple sequence alignment indicates that XG is a paralog of CD99 (Figure 3). CD99 orthologs were identified throughout the Vertebrata subphylum of the phylum Chordata, but absent from the subphylums Tunicata and Cephalochordata, suggesting that CD99 arose in the last common ancestor of the vertebrata. XG was absent from fishes and amphibians and only identified in members of the amniote clade. This suggests that XG arose by the duplication of the CD99 gene in the last common ancestor of the amniotes.

Synteny analysis shows conservation of the gene order between the CD99 chromosomal regions of zebrafish, chicken, and humans (Figure 4). The region has been enlarged in the chicken and human chromosome by the insertion of other genes (Figure 5). This analysis also reveals that CD99 and XG are on the X chromosome in eutherians but on autosomes in other organisms including monotremes and marsupials. In humans, XG is located at the boundary between pseudoautosomal region 1 and X-specific sequences, while in other eutherians, XG is completely within the pseudoautosomal region [36]. It appears as if the CD99 and XG genes were within a large region that remained autosomal in non-eutherians but were incorporated into sex-chromosomes within the last common ancestor of eutherians.

## 4. Conclusions

Research with human CD99 shows that this molecule is involved with multiple critical biological processes, though the mechanism of action is still not clear. It mediates diverse cellular processes which impact immune function, inflammation, and cancer metastasis [1]. The impact of CD99 (alloantigen D) on traits of chickens is unknown. However, significant CD99 allele frequency differences are found in two sets of chicken lines that have been divergently selected for the antibody response to immunization with sheep red blood cells which include the HAS/LAS pair of lines mentioned herein and the “high” and “low” antibody lines described by Parmentier [37]. The HAS/LAS lines are known to have different responses against certain pathogens. The HAS line was more resistant to Newcastle disease virus, marble spleen disease virus, Marek’s disease virus, Eimeria tenella, Mycoplasma gallisepticum, as well as feather mites. Conversely, LAS chickens were more resistant to Staphylococcus aureus and Escherichia coli compared with HAS chickens [16,38,39]. The observation that two sets of divergently selected lines also showed differences in CD99 haplotype frequencies implies an association with antibody response. Furthermore, certain growth traits favored the LAS line. Females from LAS had higher weight early in life, began egg production earlier, and produced more eggs compared with HAS birds. Again, the alloantigen D system differences may have contributed to these effects [15]. With the development of rapid SNP-based tests to identify variants within CD99, the impact of these variants on various traits can now be studied.

It is worth noting that neither of the GWAS analyses identified the correct candidate gene. For the 600 K SNP data set, the top SNP (*p* = 2.08 × 10^−11^) is located 325 Kbp away from the start of CD99. The top SNP within the 54 K SNP data set (*p* = 8.24 × 10^−11^) was within the ARSD gene, located 104 Kbp away from CD99. Thus, while the GWAS was able to greatly narrow down the genomic region, it could not identify the actual gene. This is due to insufficient SNP in the region of interest, lack of segregation of those SNP in lines of interest, or lack of association of the SNP (i.e., within non-coding region) with the phenotype. The information from the sequence samples, and use of the candidate gene SNP panel were necessary to unequivocally identify the candidate gene. This shows the SNP chip strength in identifying specific regions, but also shows its limitation in the identification of the correct candidate gene without having additional SNP information from other sources.

In the human, the CD99 protein is known to be critical for the leukocyte transendothelial migration signaling pathway [40], thus playing an important role in leukocyte recruitment into inflamed areas [41]. Significantly different CD99 frequencies within two independent sets of chicken lines divergently selected for antibody response suggests that CD99 is also an important component of the chicken immune response. Further studies specifically examining the impact of CD99 variants are needed to confirm the impact on immune response and the health of the birds. Specific allelic combinations may generate positive or negative consequences which can be modulated through selection.

The thirteen known chicken blood systems were all initially identified with alloantisera, which was often very specific to the lines within which it was produced. The chicken B blood system identifies variations within the chicken MHC, which contains genes with major impacts on the immune system and subsequent response to multiple diseases [10]. It was not until the availability of the genome sequence and the development of DNA-based tests that extensive MHC variability could be detected in multiple chicken breeds, including indigenous populations and wild birds [42,43,44,45]. The genes responsible for the chicken A and E blood systems have been recently identified as C4BPM and FCAMR, which are either within or very close to the Regulators of Complement Activation (RCA) region [11]. Variation within genes of the RCA was initially identified as human blood types (Cromer and Knopps blood systems) [46,47,48], which are known to impact disease outcomes [49]. The gene responsible for a fourth chicken blood system has now been identified as CD99. In humans, the CD99 molecule, in combination with the Xg molecule, defines the human CD99/Xg^a^ blood system [50]. Thus, the four chicken blood groups for which the responsible genes have been identified all have an equivalent blood system identified in humans. It will be interesting to determine if this trend will continue when the other nine chicken blood systems are defined.

## Figures and Tables

**Figure 1 genes-14-00402-f001:**
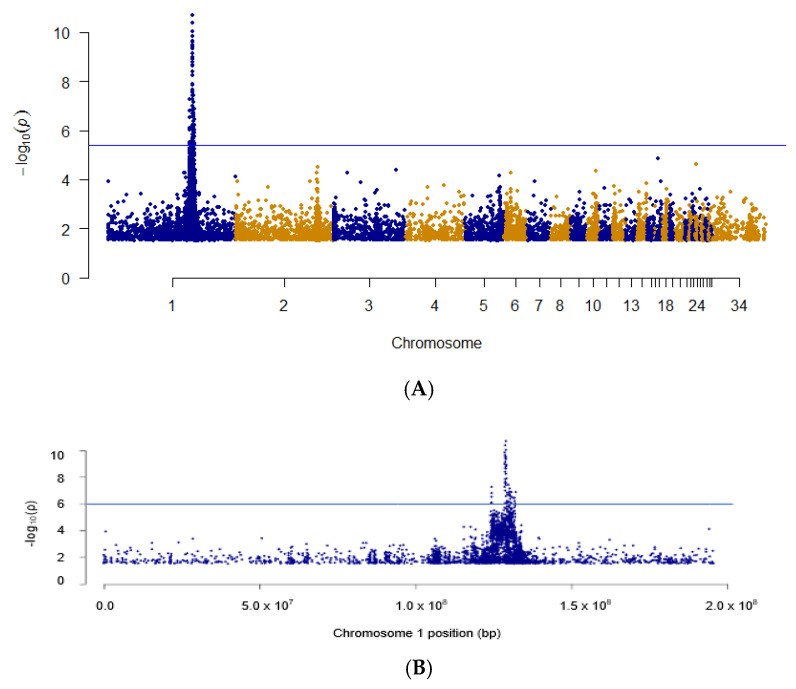
(**A**) GWAS obtained from 600 K SNP chip genotypes of pooled DNA from NIU pedigree samples segregating for D^1^ and D^2^ alleles. (**B**) Magnification of significant region on chromosome 1.

**Figure 2 genes-14-00402-f002:**
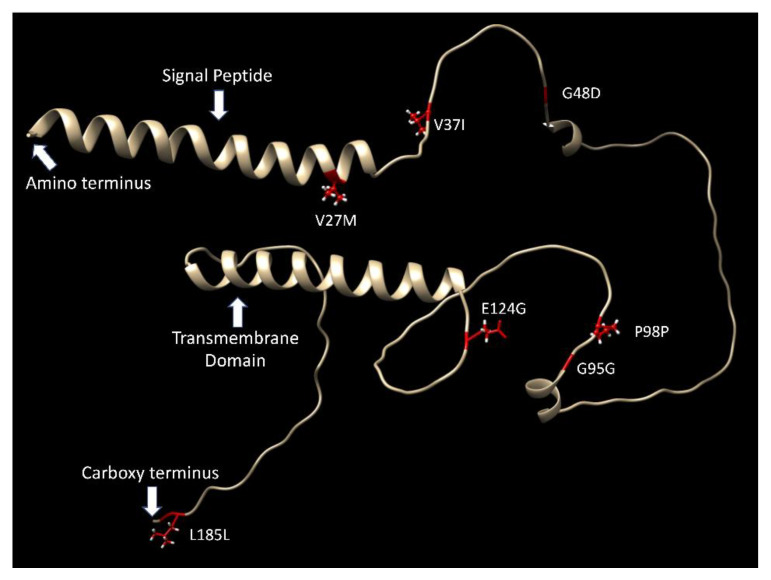
AlphaFold prediction for the structure of chicken CD99 (Uniprot E1CRC1). The transmembrane domains and signal peptide are predicted to be α helices. Other than two short α helices in the extracellular domain, the bulk of the structure is predicted to be random coil.

**Figure 3 genes-14-00402-f003:**
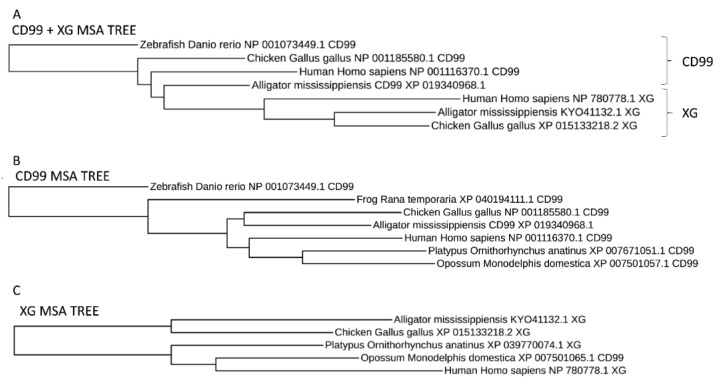
Phylogenetic trees for (**A**) CD99 and XG combined, (**B**) CD99 only, and (**C**) XG only. CD99 and XG are paralogous and XG appears to have arisen in the last common ancestor to amniotes (see text).

**Figure 4 genes-14-00402-f004:**
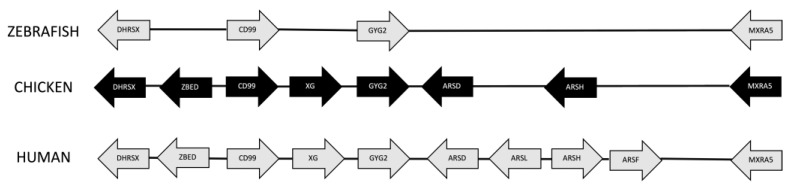
Syntenic relationships between zebrafish (Chr: 1, build: GRCz11), chicken (Chr: 1, build: GRCg7b), and human (Chr: X, build GRCh38.p14) for the CD99 chromosomal region.

**Figure 5 genes-14-00402-f005:**
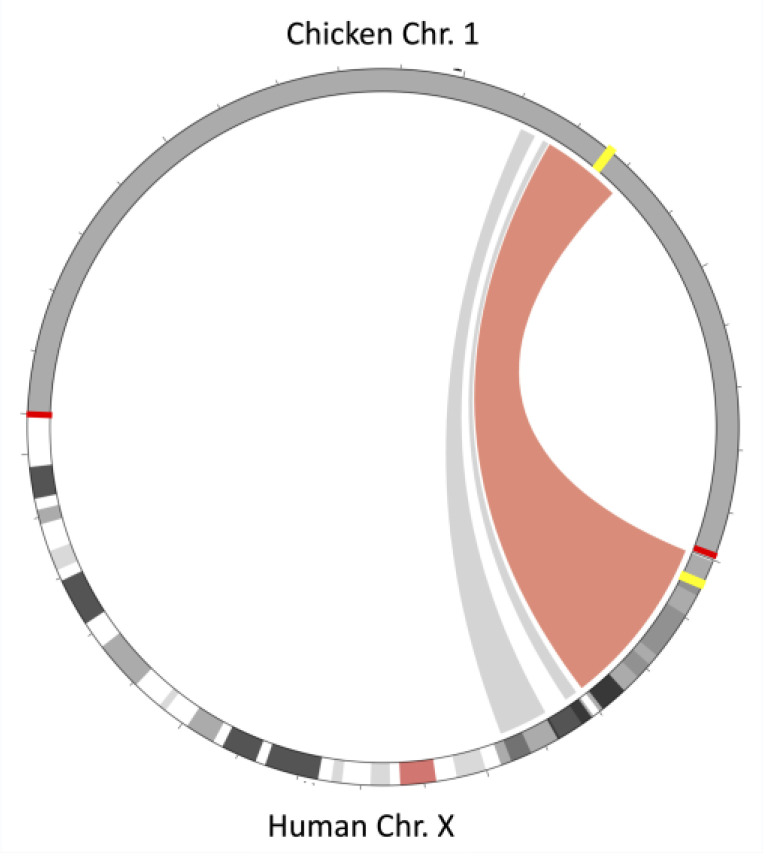
Syntenic relationship between Chicken chromosome 1 and the Human X chromosome. The brown and gray colors spanning between the Chicken to the Human chromosomes depict a region of synteny between the two species. The red bars indicate the division between the chicken and human chromosomes. In chickens, the syntenic regions span from approximately 110,900,000 to 130,000,000 on chromosome 1. In humans, the region spans from approximately 333,000 to 45,100,000 on chromosome X. The yellow bars indicate the approximate position of the CD99 and XG genes on the respective chromosomes.

**Table 1 genes-14-00402-t001:** Sources of alloantigen system *D* serologically defined samples used for candidate gene identification.

Source	Breed ^1^	D alleles present	Samples (*n*)	Reference ^2^
NIU DNA Bank; Pedigree	WL, Ancona	D^1^, D^2^	53	none
NIU DNA Bank	WL, Ancona	D^1^, D^2^, D^3^	51	none
HAS/LAS	WL	D^1^, D^3^, D^4^	44	[15]
UCD001	RJF	unk	10	[17] ^3^
UCD003	WL	D^3^	20	[19,20]
WL1	WL	D^2^, D^3^	58	none
WL2	WL	D^3^	90	none
WL3	WL	D^1^, D^3^	120	none
WL4	WL	D^1^, D^3^	119	none
WL5	WL	D^3^	120	none
RIR1	RIR	unk	64	none
WPR1	WPR	unk	320	none
WPR1	WPR	unk	272	none
Sequence only			Avg. coverage	Source
UCD001	RJF	unk	6.6	[21]
UCD003	WL	D^3^	18.7	Hans Cheng, pers. comm.
IAH-6_1_	WL	D^3^	15.2	[18]
IAH-7_2_	WL	D^1^	17.6	[18]
IAH-RHC	WL	D^3^	15.9	[18]

^1^ WL = White Leghorn, RJF = Red Jungle Fowl, RIR = Rhode Island Red, WPR = White Plymouth Rock. ^2^ Reference source for D system type information. ^3^ UCD 001 types not identified but described as different from the types in UCD 003 (D3). unk = no serology data.

**Table 2 genes-14-00402-t002:** Title SNP within CD99 gene used to define haplotypes, the SNP genomic and exon location, their predicted amino acid changes, and all SNP combinations (haplotypes) found. The D blood group serological allele found for each haplotype is given, where known.

								Haplotype										
SNP Name	Location (bp) *	Exon Gene Location	Codon Change	Nucleotide Change	aa	Ref	Alt	H01	H02	H03	H04	H05	H06 ^A^	H07	H08	H09	H10	H11
rs74153692	129,875,094	2	GTG > ATG	G > A	V27M	G	A	G	G	G	G	G	G	A	G	G	G	G
rs314496839	129,871,024	3	GTT > ATT	G > A	V37I	G	A	A	G	G	G	A	G	G	G	A	G	A
rs312634736	129,870,489	4	GGC > GAC	G > A	G48D	G	A	G	A	G	G	A	G	G	G	A	G	G
rs316308207	129,867,634	7	GGT > GGC	T > C	G95G	T	C	C	C	C	C	C	T	C	T	T	T	C
rs10730300	129,867,625	7	CCA > CCG	A > G	P98P	A	G	A	A	G	A	A	A	A	A	A	A	A
rs735363747	129,866,188	8	GAG > GGG	A > G	E124G	A	G	A	A	A	A	A	A	A	A	A	A	A
rs13623448	129,866,175	intron			splice region	ref	del	del	del	ref	ref	ref	ref	del	del	del	ref	del
rs735519530	129,864,266	11	TTG > CTG	T > C	L185L	T	C	T	T	T	T	T	T	T	T	T	T	C
D blood system serological allele								D^1^	D^2^	D^3^	D^3^	unk	D^3^	unk	unk	unk	unk	unk

Exons and amino acid numbering as defined by ENSGALT00000041401; * location based on GRCg6a; Ref = reference sequence; Del = deleted; ^A^ haplotype found in the chicken Junglefowl reference genome; unk = unknown.

**Table 3 genes-14-00402-t003:** Alloantigen system D alleles, their corresponding CD99 haplotypes, and sources of each allele.

A Allele	Haplotype	Source(s)
D^1^	CD99-H01	NIU DNA Bank and NIU pedigree, IAH-7_2_, HAS, LAS, WL4, WL3
D^2^	CD99-H02	NIU DNA Bank and NIU pedigree, WL1
D^3^	CD99-H03	NIU DNA Bank, IAH-6_1_, IAH-RHC, UCD003, HAS, LAS, WL1, WL2, WL3, WL4, WL5
D^3^	CD99-H04	NIU DNA Bank
D^3^	CD99-H06	HAS, WL3

## Data Availability

The data presented in this study are available on request from the corresponding author. The data are not publicly available due to proprietary content from Hy-Line International.

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
