# Peer review of "CD99 and the Chicken Alloantigen D Blood Systemâ€"

_genes, 2023, doi:10.3390/genes14020402_

Round 1
Reviewer 1 Report
In the manuscript entitled “CD99 and the Chicken alloantigen D blood system”, Fulton et al. present the identification of the gene involved in the chicken D blood system. The identification of this gene is a progress because only the approximate region of the genome was known until now. To identify the gene, they combined SNP genotyping and genome sequences and the approach sounds correct. However, the analysis of the context of the gene and its evolution must be thoroughly improved. The scope of the study is not really emphasized in the text. More background and discussion of the chicken blood system seems necessary.
Major comments
- Introduction: More context is needed on the chicken D blood system, its role and importance.
- Structure:
Figure 2. Why do the authors show the predicted structure of the human protein when we are interested in the chicken protein (which is also available from AlphaFold)? This needs to be changed.
The position of the mutated amino acids needs to be further discussed (intracellular, extracellular...).
- Protein family analysis:
As it stands, the analysis is superficial to say the least. Sequences from more model species and the CD99 antigen-like protein 2 paralogous family should be included. One cannot make conclusions by leaving out a whole part of the evolutionary history of the family. At least one event of duplication is ignored here. The multiple alignment must be made available (at least in the supplementary materials) with the access numbers of the protein sequences and the position of the different features of these families (signal peptide, extracellular domain...). This alignment should be used to build a single tree illustrating the evolution of the complete family. The name of the phylogenetic program must be indicated. Bootstrap or other values should be calculated to assess the robustness of each node. The tree rooting should be justified.
- Discussion:
The value of having identified alloantigen D should be further emphasized and put into perspective. The authors insist on the synteny of the CD99 and Xg genes but do not discuss it at all.
Minor comments:
- L38 : “The CD99 protein together with the highly homologous Xg antigen”: “highly homologous” means nothing, 2 genes are homologous or not. “With its highly conserved homolog” would be correct.
L41 “Xg and CD99 are homologous genes” : This has already been stated just before. It would be better to simply say "Human and CD99 genes, each consisting of 10 exons, encode...”
L41: The Xg gene has 11 exons, at least for the Ensembl canonical transcript.
L43: The acronym RBC should be defined when first used.
L44 : Rephrase the sentence.
L132: “Biomart [21] and Uniprot databases 132 [22] were used to identify cellular components for all genes.” Replace genes by proteins.
L136: Same remark, replace gene by protein.
Figure 4: the chromosome number and version of assemblies should be indicated. For human, the orientation of the ARSF gene is wrong. Some ncRNA genes are indicated, others are lacking. The order and respective orientation of genes don’t correspond to the GRCz11 assembly of zebrafish.
Figure 5: This appears to be a low definition screenshot that may not belong in the main text.
Author Response
In the manuscript entitled “CD99 and the Chicken alloantigen D blood system”, Fulton et al. present the identification of the gene involved in the chicken D blood system. The identification of this gene is a progress because only the approximate region of the genome was known until now. To identify the gene, they combined SNP genotyping and genome sequences and the approach sounds correct. However, the analysis of the context of the gene and its evolution must be thoroughly improved. The scope of the study is not really emphasized in the text. More background and discussion of the chicken blood system seems necessary.
The primary purpose of this study was to identify the gene responsible for the D blood system. The analysis of the gene and its evolution was included to provide additional information and relevance, but was not intended to be the main focus. Very little is known about the various chicken blood group systems, as little has been done on them since their initial discovery. This is the only manuscript with any type of molecular information on the D blood system making it a unique opportunity for Genes. Modifications have been made in the text to provide better explanation and emphasis.
Major comments
- Introduction: More context is needed on the chicken D blood system, its role and importance.
More information has been added, but very little is known about the impact of the D system alleles on the chicken. Now that the gene had been identified, and DNA based tests developed to detect the various alleles (as reported in this manuscript), additional work can be pursued.
- Structure:
Figure 2. Why do the authors show the predicted structure of the human protein when we are interested in the chicken protein (which is also available from AlphaFold)? This needs to be changed. We apologize, the structure provided was actually the chicken, not the human. We have added the Uniprot ID for the structure used. E1BRC1. The human and chicken structures are almost identical.
The position of the mutated amino acids needs to be further discussed (intracellular, extracellular...). We did indicate that with the exception of one of the SNP all others are located on the random coil. Additional information has been provided.
- Protein family analysis:
As it stands, the analysis is superficial to say the least. Sequences from more model species and the CD99 antigen-like protein 2 paralogous family should be included. One cannot make conclusions by leaving out a whole part of the evolutionary history of the family. At least one event of duplication is ignored here. The multiple alignment must be made available (at least in the supplementary materials) with the access numbers of the protein sequences and the position of the different features of these families (signal peptide, extracellular domain...). This alignment should be used to build a single tree illustrating the evolution of the complete family. The name of the phylogenetic program must be indicated. Bootstrap or other values should be calculated to assess the robustness of each node. The tree rooting should be justified. The primary purpose of this manuscript was to identify the D blood system candidate gene. The information on the protein family analysis was added primarily as additional interest for the reader to tie together This protein family analysis section has been removed.
- Discussion:
The value of having identified alloantigen D should be further emphasized and put into perspective. The authors insist on the synteny of the CD99 and Xg genes but do not discuss it at all. The synteny of CD99 and Xg is well known for the human. The comments on the protein synteny of CD99 and Xg has been removed, thus this no longer needs to be added to the discussion.
Minor comments:
- L38 : “The CD99 protein together with the highly homologous Xg antigen”: “highly homologous” means nothing, 2 genes are homologous or not. “With its highly conserved homolog” would be correct. done
L41 “Xg and CD99 are homologous genes” : This has already been stated just before. It would be better to simply say "Human and CD99 genes, each consisting of 10 exons, encode...” done
L41: The Xg gene has 11 exons, at least for the Ensembl canonical transcript. Multiple spicing can result in 10 or 11 exons. done
L43: The acronym RBC should be defined when first used. done
L44 : Rephrase the sentence. DONE Changed to “In humans, the CD99 gene lies within pseudoautosomal region 1,, while part of the Xg gene is located within the pseudoautosomal region and a portion is X specific”.
L132: “Biomart [21] and Uniprot databases 132 [22] were used to identify cellular components for all genes.” Replace genes by proteins. done
L136: Same remark, replace gene by protein. done
Figure 4: the chromosome number and version of assemblies should be indicated. For human, the orientation of the ARSF gene is wrong. Some ncRNA genes are indicated, others are lacking. The order and respective orientation of genes don’t correspond to the GRCz11 assembly of zebrafish. We thank the reviewer for noticing the orientation errors, and we have now corrected the gene orientations. The ncRNA was removed as we wish to focus on coding genes. The chromosome number and assembly versions used are added.
Figure 5: This appears to be a low definition screenshot that may not belong in the main text.
We included this figure to show the extent of synteny between the chicken chr 1 and the human chr x. We have added additional information to the figure and improved the definition as we still think this is a valid method of showing the synteny and is of interest to the reader. Resolution of the figure has been improved.
Reviewer 2 Report
This manuscript examines the genetic origin of D blood group in chickens (Gallus gallus). The research methodology is well designed and the authors have tested their idea using various verification methods. This paper is organized by expert poultry geneticists and is likely to receive many citations. Moreover, this article will be useful for the readers of genes magazine. Some minor corrections are recommended in the attached file.

Author Response
This manuscript examines the genetic origin of D blood group in chickens (Gallus gallus). The research methodology is well designed and the authors have tested their idea using various verification methods. This paper is organized by expert poultry geneticists and is likely to receive many citations. Moreover, this article will be useful for the readers of genes magazine. Some minor corrections are recommended in the attached file. The double spaces at the beginning of a sentence have been removed. Other minor changes as requested have been done
Reviewer 3 Report
As we all known, the chicken D blood system is one of 13 alloantigen systems found on chicken red blood 12 cells. The author found that chicken CD99 gene showed co-segregation 20 of SNP defined haplotypes and serologically defined D blood system alleles. The CD99 protein mediates multiple cellular processes including leukocyte migration, T-cell adhesion, affecting peripheral immune responses. However, this manuscript was full of data analysis, which lack of sufficient experiment. We hence ask the author to perform several related experiment to convince us the function of Chicken CD99. Without those experiment, we could not accept this manuscript in this time.
Author Response
As we all known, the chicken D blood system is one of 13 alloantigen systems found on chicken red blood 12 cells. The author found that chicken CD99 gene showed co-segregation 20 of SNP defined haplotypes and serologically defined D blood system alleles. The CD99 protein mediates multiple cellular processes including leukocyte migration, T-cell adhesion, affecting peripheral immune responses. However, this manuscript was full of data analysis, which lack of sufficient experiment. We hence ask the author to perform several related experiment to convince us the function of Chicken CD99. Without those experiment, we could not accept this manuscript in this time.
The primary purpose of this manuscript was to identify the candidate gene responsible for the chicken D blood system, and report DNA based testing methods of defining the different alleles. The observation of the impact of the D system on various phenotypic traits as reported in the literature was added to provide relevance of this variation. The impact of CD99 in humans is well reported and of significant impact to the immune system. This adds value to the identification of the chicken D blood group as being due to CD99. There are no studies with chickens on CD99 function. CD99 functional experiments are beyond the scop of this manuscript. This can be done in future work now that we have a rapid method to identify variation in the responsible gene.
Round 2
Reviewer 1 Report
The authors adressed all my comments.
Reviewer 3 Report
Since no further experiment related to CD99 were performed, we are very sorry for that we recommend rejecting this manuscript in this time.